# Towards Accurate Subgraph Similarity Computation via Neural Graph Pruning

**Linfeng Liu** [*]                                              *linfengliu@meta.com*
*Meta, Boston*

**Xu Han**                                                       *Xu.Han@tufts.edu*
*Department of Computer Science*
*Tufts University*

**Dawei Zhou**                                                   *zhoud@vt.edu*
*Department of Computer Science*
*Virginia Tech*

**Li-Ping Liu**                                                  *liping.liu@tufts.edu*
*Department of Computer Science*
*Tufts University*

**Reviewed on OpenReview:** *https://openreview.net/forum?id=CfzIsWWBlo*

## Abstract

Subgraph similarity search, one of the core problems in graph search, concerns whether a target graph approximately contains a query graph. The problem is recently touched by neural methods. However, current neural methods do not consider pruning the target graph, though pruning is critically important in traditional calculations of subgraph similarities. One obstacle to applying pruning in neural methods is the discrete property of pruning. In this work, we convert graph pruning to a problem of node relabeling and then relax it to a differentiable problem. Based on this idea, we further design a novel neural network to approximate a type of subgraph distance: the subgraph edit distance (SED). In particular, we construct the pruning component using a neural structure, and the entire model can be optimized end-to-end. In the design of the model, we propose an attention mechanism to leverage the information about the query graph and guide the pruning of the target graph. Moreover, we develop a multi-head pruning strategy such that the model can better explore multiple ways of pruning the target graph. The proposed model establishes new state-of-the-art results across seven benchmark datasets. Extensive analysis of the model indicates that the proposed model can reasonably prune the target graph for SED computation. The implementation of our algorithm is released at our Github repo: `https://github.com/tufts-ml/Prune4SED`.

## 1 Introduction

Graphs are important tools for describing structured and relational data (Wu et al., 2020b) from small molecules and large social networks. Among fundamental operations in graph analysis, the calculation of subgraph similarity is one important problem (Yan et al., 2005). Subgraph similarity search concerns whether a target graph approximately contains a query graph (Samanvi & Sivadasan, 2015; Shang et al., 2010; Peng et al., 2014; Zhu et al., 2012; Yuan et al., 2012). Subgraph similarity search has a wide range of applications, which span over various fields, including drug discovery (Ranu et al., 2011), computer vision (Petrakis & Faloutsos, 1997), social networks (Samanvi & Sivadasan, 2015), and software engineering (Wu et al., 2020a).

---

[*]Work done while at Tufts University.

There are multiple ways of defining the similarity or distance[1] between a query graph and a target graph. This work focuses on Subgraph Edit Distance (SED) (Riesen, 2015), which is a type of pseudo-distance from the query graph to the target graph (Bunke, 1997; Bougleux et al., 2017; Zeng et al., 2009; Fankhauser et al., 2011; Daller et al., 2018; Riesen & Bunke, 2009). Concretely speaking, SED is the minimum number of edits that transform the query graph to a subgraph of the target graph. Such edits include insertion of nodes or edges, deletion of nodes or edges, and substitution of nodes or edge labels. When SED is zero, then the query graph is isomorphic to a subgraph of the target graph.

Exact computation of SED is NP-hard, which is from the fact that SED is a generalization of an NP-complete problem: the subgraph isomorphism problem. The core problem of SED calculation is to identify the subgraph in the target graph that best matches the query graph. In this type of problem, pruning plays an important role. Traditional algorithms often use a pruning procedure to remove from the target graph those nodes that are unlikely to match any nodes in the query graph. An effective pruning procedure can greatly reduce the space of the following searching procedure and thus improve both the speed and accuracy of similarity calculation. Improving the effectiveness of pruning algorithms is not an easy problem and is still a hot topic in the research of traditional algorithms (Lee et al., 2010b).

In this work, we consider neural methods for graph pruning. The main difficulty here is that graph pruning usually consists of discrete operations and is not differentiable. We overcome the difficulty by converting graph pruning to a node relabeling problem and then relaxing it to a continuous problem. This novel formulation enables the possibility of fitting a pruning model with neural networks.

We further design a new learning model, Neural Graph Pruning for SED (Prune4SED), to *learn* SEDs of graph pairs. The model first uses a *query-aware* learning component to compute node representations for the target graph such that these representations also contain information about the query graph. Then the model considers multiple prunings of the target graph using a *multi-head* structure. Finally the model compute the SED between the query and the pruned target graph. The entire model can be trained end-to-end.

The empirical study shows that Prune4SED establishes new state-of-the-art results across seven benchmark datasets. In particular, Prune4SED achieves an average of 23% improvement per dataset over the previous neural model. We also demonstrate promising results for an application of molecular fragment containment search in drug discovery: our model can accurately retrieve molecules containing given functional groups. Finally, extensive analysis of our model confirms that it can effectively prune a significant fraction of nodes that cannot match the query graph.

## 2 Related Work

Based on edit distance (Bougleux et al., 2017; Zeng et al., 2009; Fankhauser et al., 2011; Riesen, 2015; Daller et al., 2018; Riesen & Bunke, 2009), SED is an expressive form to quantify subgraph similarity. Exact computation of SED is often infeasible because it is NP-hard. Pruning is an effective strategy in the detection of query graphs from target graphs (Lee et al., 2010b).

Graph Neural Networks (GNNs) (Wu et al., 2020b) apply deep learning models to learn from graph-structured data. GNNs have been tested powerful to embed a graph structures into vector representations. (Hamilton et al., 2017; Veličković et al., 2017; Xu et al., 2018; Liu et al., 2020; Chen et al., 2020). Though standard message-passing GNNs (Chen et al., 2020) has limitations in identifying subgraphs, various remedies (Sato, 2020) help to overcome these problems.

Recent advancements in GNNs open the possibility of neural combinatorial optimization (Cappart et al., 2021). For example, GNNs have been used to count isomorphic subgraphs (Liu et al., 2020), graph matching (Liu et al., 2021), and subgraph matching (Lou et al., 2020). Recently GNNs are applied to compute edit distance (Li et al., 2019; Bai et al., 2019; 2020; Zhang et al., 2021). The most relevant work is NeuroSED (Ranjan et al., 2021), which learns to predict SED via GNNs. While the method shows promising results for SED calculations, it faces difficulties when the target graph is much larger than the query: it becomes

---

[1]Here "distance" is a loose term and but not a "metric" that has a strict definition in math.

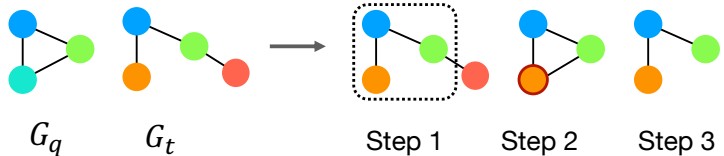

Figure 1: An example of computing SED between a query graph $G_q$ and a target graph $G_t$. The first step is to find an optimal subgraph in $G_t$. The last two steps are substituting a node label and deleting an edge in $G_q$, ending up SED as 2.

harder for the model to identify the subgraph that is most similar to the query. This issue motivates us to explicitly consider pruning in a neural model.

For applications such as graph matching where the input is a graph pair, it is beneficial to use cross-graph information in GNNs. The main idea is to allow information flow during GNN's propagation (Li et al., 2019; Ling et al., 2020; 2021; Wang et al., 2019). For example, after a GNN propagation, node representation in one graph will be further updated by using node representation from the other graph (Li et al., 2019).

## 3 Preliminaries

A graph $G = (V, E, X)$ consists of a node set $V$ and an edge set $E$. $X(i)$ represents the label of a node $i \in V$; let $\mathbf{x}_i = \text{ONEHOT}(X(i))$ be the one-hot encoding of $i$'s node label; and let $\mathbf{X} = (\mathbf{x}_i)_{i \in V}$ denote the node label matrix, whose rows are one-hot encodings of node labels. Let $\Sigma$ be the universe of all node labels. We do not consider graphs with edge labels for now but will discuss extensions to include such cases.

**Subgraph Edit Distance (SED).** Given a query graph $G_q = (V_q, E_q, X_q)$ and a target graph $G_t = (V_t, E_t, X_t)$, SED represents the minimum cost of edits on $G_q$ such that the edited query is isomorphic to a subgraph in $G_t$. An edit operation can be inserting a node or an edge, deleting a node or an edge, or modifying a node or edge label (Zeng et al., 2009). Alternatively, the computation of SED seeks to identify a subgraph $G_s$ in the target graph $G_t$ such that $G_s$ and $G_q$ have the minimum graph edit distance (GED), which is the minimum number of edits that convert $G_q$ to be isomorphic $G'_t$.

$$\text{SED}(G_t, G_q) = \min_{G_s \subseteq G_t} \text{GED}(G_s, G_q). \tag{1}$$

Here we only consider connected graphs $G_t$ and $G_q$. Figure 1 illustrates an example of SED computation. In a geneal version of SED, each type of edit operation has a cost, which is application-dependent. Here we consider the basic version and set 1 as cost for all types of edits (e.g. (Zheng et al., 2013; Bai et al., 2019)).

**Graph Neural Networks (GNNs).** GNNs learn node representations by iteratively updating node representations and sending messages to neighbors.

$$\mathbf{h}'_i = \text{UPDATE}\left(\mathbf{h}_i, \text{AGGREGATE}\left(\{\mathbf{h}_j : (i, j) \in E\}\right)\right). \tag{2}$$

Here UPDATE denotes an update function and AGGREGATE denotes an aggregation function. For example, AGGREGATE sums up all input vectors, and UPDATE is a dense layer that applies to the concatenation of its two arguments. $\mathbf{h}_i$ denotes the current representation of $i$, and $\mathbf{h}'_i$ denotes updated representation. GNNs capture node features and topological features of a graph simultaneously (Wu et al., 2020b). GNN variants use different UPDATE and AGGREGATE functions. This work uses GATv2Conv (Brody et al., 2021), but other GNN alternatives such as Papp et al. (2021); Zhang & Li (2021) can also be considered. The choice of GNN architectures is a model selection problem , and we leave such exploration to the future.

A pooling function is often applied to aggregate all node vectors into a single graph vector.

$$\mathbf{z} = \text{POOL}\left(\{\mathbf{h}_i : i \in V\}\right) \tag{3}$$

The POOL function is invariant to the order of elements in its input and has several implementations. For example, the POOL function can be an average function followed by a dense layer (Hamilton et al., 2017).

**NeuroSED.** NeuroSED (Ranjan et al., 2021) is a neural network designed to mimic the calculation of SEDs from graph pairs. It uses Graph Isomorphism Network (GIN) (Xu et al., 2018) to encode both $G_t$ and $G_q$ into vectors and then predicts SED using a simple fixed function. Formally, NeuroSED writes as:

$$\mathbf{H}_t = \text{GIN}(G_t), \quad \mathbf{H}_q = \text{GIN}(G_q), \tag{4}$$

$$\text{SED} = \|\text{ReLU}\left(\text{POOL}(\mathbf{H}_q) - \text{POOL}(\mathbf{H}_t)\right)\|_2. \tag{5}$$

Here $\|\cdot\|_2$ is the $L_2$ norm.

## 4 Neural Graph Pruning for SED Calculation

### 4.1 Graph Pruning as a Node Relabeling Problem

Graph pruning aims to remove nodes in the target graph without affecting the SED between the target graph and the query graph. Suppose graph pruning keeps a node subset $V_t' \subset V_t$ of the target graph and get the induced subgraph $G_t' = (V_t', E_t', X_t(V_t'))$. The purpose of pruning is to maintain that

$$SED(G_t, G_q) \approx SED(G_t', G_q) \tag{6}$$

while maximizing the number $|V_t| - |V_t'|$ of pruned nodes. Graph pruning is a discrete operation, and the solution $V_t'$ exists in a large searching space.

Here we formulate graph pruning as a node relabeling problem, which is more convenient for learning models.

We use a new node label $\phi \notin \Sigma$ to indicate that a node is pruned. Specifically, we relabel the target graph by $X_t''$ such that

$$X_t''(i) = \begin{cases} X_t(i) & \text{if } i \in V_t' \\ \phi & \text{otherwise} \end{cases} \tag{7}$$

We further assign a large cost (e.g. the size of $G_t$) to the edit operation that switches an actual label with $\phi$. Since the cost exceeds the edit distance from $G_q$ to any subgraph of $G_t$, the new graph $G_t'' = (V_t, E_t, X_t'')$ with relabeling is equivalent to the pruned graph $G'$ in terms of SED calculation:

$$SED(G_t', G_q) = SED(G_t'', G_q). \tag{8}$$

This is because the latter calculation will not match any node outside of $V_t'$ to a node in $G_q$; otherwise, a node outside of $V_t'$ will incur a large edit distance due to its new label $\phi$. A target graph with relabeling is shown in Figure 2.

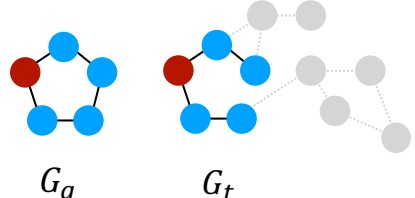

Figure 2: Graph pruning in SED as a node relabeling problem. Gray nodes, which take the special label $\phi$, are treated as pruned.

To incorporate graph pruning in a neural network, we further formulate node relabeling to a continuous problem. In a neural network, the label of a node $i$ is usually expressed by a one-hot vector $\mathbf{x}_i = \text{ONEHOT}(X_t(i))$, then we relabel each node $i$ by

$$\mathbf{x}_i' = \alpha_i \mathbf{x}_i. \tag{9}$$

Here $\alpha_i$ is binary. If $\alpha_i = 0$, which indicates the pruning of node $i$, then $\mathbf{x}_i'$ is a zero vector, which is different from any one-hot representation of original labels in $\Sigma$. The vector $\boldsymbol{\alpha} = (\alpha_i : i \in V_t)$ is a binary vector indicating which nodes are kept.

Then we relax the indicator vector to be continuous, $\boldsymbol{\alpha} \in [0, 1]^{|V_t|}$. We call the continuous vector as *keep probability*: nodes with small probabilities are considered to be "pruned" from the graph. Now a neural network can compute $\boldsymbol{\alpha}$ from $G_t$ and $G_q$ as a learnable pruning component for neural SED calculation.

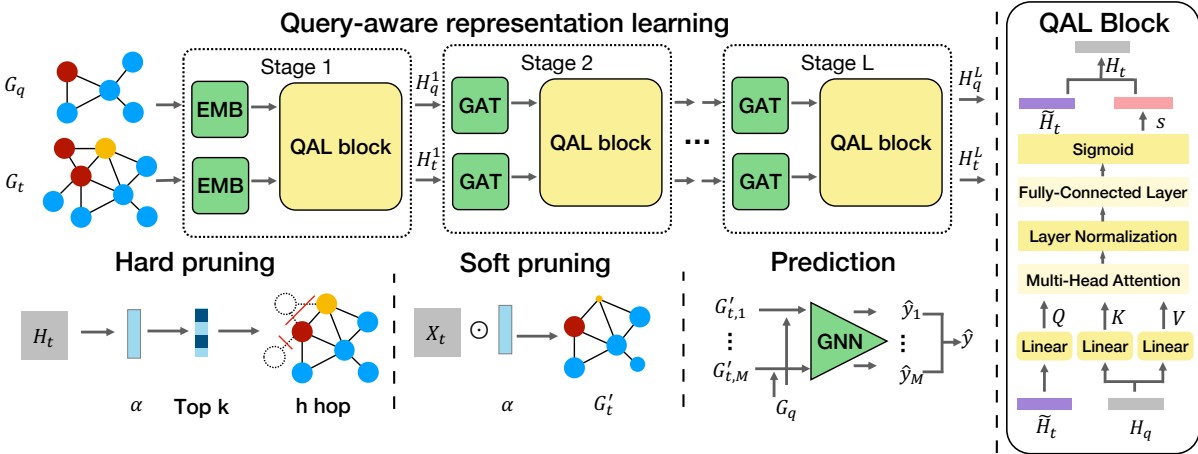

Figure 3: The overall framework of Prune4SED, including query-aware representation learning, hard/soft pruning, and multi-head SED prediction. The query-aware learning (QAL) block is implemented by a multi-head attention layer.

In the neural model, there is no need to specify the editing cost between a zero vector and a one-hot vector because the model can automatically decide it when fitting the training data. If $\boldsymbol{\alpha}$ reflexes accurate pruning, then a flexible learning model would avoid using nodes with zero vectors to predict accurate SED values.

## 4.2 Graph Pruning for SED Calculation

With the formulation above, we design an end-to-end neural model for computing SED. An overview of the model is depicted in Figure 3 (left). Specifically, Prune4SED has three components, i.e., (1) query-aware representation learning, (2) hard/soft pruning, and (3) multi-head SED predicting. The first component learns node representation for $G_t$ conditioned on $G_q$. The second component learns to keep probability $\boldsymbol{\alpha}$ and also does the pruning. The third component predicts the final SED value.

**Query-aware representation learning.** The representation learning component takes in the target $G_t$ and the query $G_q$ and outputs node representations of $G_t$. We design a neural architecture as the component to effectively use the information about $G_q$ when calculating node representations for $G_t$.

The core part of the component is the query-aware learning (QAL) block, which allows the information of $G_q$ to flow into node representations of $G_t$. Figure 3 (right) shows the structure of QAL. The inputs to the block are node representations $\mathbf{H}_q$ of $G_q$ and node representations $\widetilde{\mathbf{H}}_t$ of $G_t$. These representations are either from node features or learned by GNNs, which we will elaborate on later. Then the QAL block updates $\widetilde{\mathbf{H}}_t$ to new representations $\mathbf{H}_t$ using attention mechanism (Vaswani et al., 2017).

$$\mathbf{H}_t = \text{QAL}\left(\widetilde{\mathbf{H}}_t, \mathbf{H}_q\right). \tag{10}$$

The function of the QAL is specified as below:

$$\mathbf{H}_t = \text{diag}(\mathbf{s})\,\widetilde{\mathbf{H}}_t, \quad \mathbf{s} = \text{sigmoid}\left(\text{MLP}(\mathbf{R})\right), \mathbf{R} = \text{MHA}\left(\mathbf{Q} = \widetilde{\mathbf{H}}_t, \mathbf{K} = \mathbf{V} = \mathbf{H}_q\right). \tag{11}$$

Here MHA is a multi-head attention layer (Vaswani et al., 2017) described in A.1. An intuitive understanding here is that representations of the target graph queries information from the query graph. Then we compute a weight vector $\mathbf{s} \in [0,1]^{|V_t| \times 1}$ that measures node importance in $G_t$. Then we scale $\widetilde{\mathbf{H}}_t$ according to $\mathbf{s}$, yielding the updated node representation $\mathbf{H}_t$ for $G_t$.

Now we are ready to compose the entire representation learning component with QAL blocks and GNN layers. We first use an embedding layer (the function $\text{EMB}(\cdot)$) to encode node labels of both $G_t$ and $G_q$ into

vector representations $\tilde{\mathbf{H}}_t^1$ and $\mathbf{H}_q^1$. Then we alternately apply QAL blocks and GNN layers to compute $G_t$ and $G_q$'s representations $\mathbf{H}_t^l$ and $\mathbf{H}_q^l$.

$$\mathbf{H}_t^1 = \text{QAL}\left(\tilde{\mathbf{H}}_t^1, \mathbf{H}_q^1\right), \quad \tilde{\mathbf{H}}_t^1 = \text{EMB}\left(\mathbf{X}_t\right), \quad \mathbf{H}_q^1 = \text{EMB}\left(\mathbf{X}_q\right), \tag{12}$$

$$\mathbf{H}_t^l = \text{QAL}\left(\tilde{\mathbf{H}}_t^l, \mathbf{H}_q^l\right), \quad \tilde{\mathbf{H}}_t^l = \text{GNN}\left(\mathbf{H}_t^{l-1}, E_t\right), \quad \mathbf{H}_q^l = \text{GNN}\left(\mathbf{H}_q^{l-1}, E_q\right), \quad l = 2, \dots, L \tag{13}$$

Here GNN is a 1-layer graph attention convolution (Brody et al., 2021) (see A.2). The embedding layer and GAT layers are shared by $G_q$ and $G_t$.

We compute the final node representations $\mathbf{H}_t$ for $G_t$ using node representations after each QAL block.

$$\mathbf{H}_t = \text{MLP}\left(\text{CONCAT}\left(\mathbf{H}_t^1 \dots, \mathbf{H}_t^L\right)\right) \tag{14}$$

Here CONCAT concatenates node representations of $G_t$ at all stages to capture multi-granular views at various granularity levels. The MLP then encodes the concatenated node representations, further distilling essential information for pruning.

The query-aware learning component learns node representations of $G_t$ in the context of $G_q$. Each QAL block checks whether a node in $G_t$ can be matched to nodes in $G_q$ by considering their respective surrounding structures. Since GAT layers encode structural information at different levels of granularity, QAL blocks consider neighborhoods also at different levels. Therefore, it is beneficial to concatenate representations learned from all QAL blocks to capture information at different levels.

**Hard/soft pruning.** Then we compute the keep probability $\boldsymbol{\alpha}$ for $G_t$ using $\mathbf{H}_t$.

We calculate $\boldsymbol{\alpha}$ from $G_t$'s node representations after each QAL block.

$$\boldsymbol{\alpha} = \text{Sigmoid}\left(\frac{\mathbf{H}_t \mathbf{p}}{\|\mathbf{p}\|}\right). \tag{15}$$

Here $\boldsymbol{\alpha}$ is computed by a scalar projection along rows of $\mathbf{H}_t$ on a learnable vector $\mathbf{p}$, then it is scaled to $(0, 1)$ by the sigmoid function. The projection is inspired by previous works in graph pooling (Gao & Ji, 2019; Cangea et al., 2018; Knyazev et al., 2019).

Next we use $\boldsymbol{\alpha}$ to prune $G_t$. Before we apply the type of pruning we discussed in equation 9, we first apply *hard pruning* to remove some nodes from $G_t$. We first keep a set $S$ of $k$ nodes corresponding to the largest value in $\boldsymbol{\alpha}$. We also keep all $h$ hop neighbors of each node in $S$ to increase connectivity and balance computation and information loss. Then we get a set $V_t'$ of nodes as the result of hard pruning from $\boldsymbol{\alpha}$. Thus, we define hard pruning as follows:

$$S = \text{top\_}k(\boldsymbol{\alpha}), \tag{16}$$
$$V_t' = h\_\text{hop}(S, G_t), \tag{17}$$

Here $\text{top\_}k(\boldsymbol{\alpha})$ selects top $k$ nodes corresponding to the $k$ largest values in $\boldsymbol{\alpha}$; and $h\_\text{hop}(S, G_t)$ extracts $h$ hop neighbors surrounding nodes in $S$.

The hard pruning is useful when $G_t$ has a much larger diameter than $G_q$. In this case, it removes a decent fraction of nodes in $G_t$, which is beneficial to both accuracy and speed. Later we show the effectiveness of hard pruning in real examples in Figure 5, . Hard pruning shares the same principle as Graph U-Net (Gao & Ji, 2019), which shows that the discrete operation does not pose issues to loss minimization.

Then we apply $\boldsymbol{\alpha}$ to these nodes in $V_t'$ to do *soft pruning* as introduced in Section 4.1.

$$X_t'(i) = \alpha_i X_t(i), \quad i \in V_t' \tag{18}$$

The pruned graph is $G_t' = (V_t', E_t', X_t')$. Here $E_t'$ keeps all $G_t$'s edges that are incident with nodes in $V_t'$.

We put the entire pruning procedure into a single function $G_t' = \text{PRUNE}(\mathbf{H}_t, G_t)$, which computes $\boldsymbol{\alpha}$ from $\mathbf{H}_t$ and then executes hard and soft pruning on $G_t$ to get $G_t'$. Note that soft pruning is differentiable and

---

**Algorithm 1** Prune4SED

    **Input:** $G_q, G_t$

1: ▷ **Query-aware representation learning**
2: **for** $l = 1$ to $L$ **do**
3:     $\widetilde{\mathbf{H}}_t^l, \mathbf{H}_q^l = \begin{cases} \text{EMB}\left(\mathbf{X}_t\right), \ \text{EMB}\left(\mathbf{X}_q\right), & \text{if } l = 1 \\ \text{GAT}\left(\mathbf{H}_t^{l-1}, E_t\right), \ \text{GAT}\left(\mathbf{H}_q^{l-1}, E_q\right), & \text{o.w.} \end{cases}$
4:     $\mathbf{H}_t^l = \text{QAL}(\widetilde{\mathbf{H}}_t^l, \mathbf{H}_q^l)$                            ▷ QAL block from equation 10
5: **end for**
6: $\mathbf{H}_t = \text{MLP}\left(\text{CONCAT}\left(\mathbf{H}_t^1 \ldots, \mathbf{H}_t^L\right)\right)$
7: **for** $m = 1$ to $M$ **do**                                     ▷ Multi-head pruning
8:     $G_{t,m}' = \text{PRUNE}_m\left(\mathbf{H}_t, G_t\right)$
9:     $\hat{y}_m = \text{PRED}(G_{t,m}', G_q)$
10: **end for**
11: $\hat{y} = \text{MEAN}\left(\hat{y}_1, \ldots, \hat{y}_M\right)$
12: **Return** $\hat{y}$

---

helps to learn the keep probability $\alpha$. The model learns to use small keep probabilities to indicate that their corresponding nodes cannot be matched to nodes in the query graph. Hard pruning is *not* differentiable, but it is still meaningful to remove nodes with small keep probabilities.

**Multi-head prediction of SEDs.** Once we have the pruned graph $G_t'$, we use a neural module to "predict" the SED between $G_t'$ and $G_q$. For example, we can use the NeuralSED network as the predictive module.

However, given the combinatorial nature of graph pruning, there might be many locally optimal solutions of $\boldsymbol{\alpha}$, and it is hard for the model to identify good $\boldsymbol{\alpha}$ values in one-shot. Inspired by multi-head attention, we propose a multi-head predicting module.

We use $M$ separate pruning functions, $(\text{PRUNE}_1, \ldots, \text{PRUNE}_M)$, each of which optimizes its own parameter $\mathbf{p}_m$ in equation 15, to obtain $M$ different pruned graphs $G_{t,1}', \ldots, G_{t,M}'$. Then we use the same predictive module PRED to compute SED values from these pruned graphs.

$$G_{t,m}' = \text{PRUNE}_m\left(\mathbf{H}_t, G_t\right), \quad \hat{y}_m = \text{PRED}(G_{t,m}', G_q), \quad m = 1, \ldots, M. \tag{19}$$

Then we take the average to get the final prediction.

$$\hat{y} = \text{MEAN}\left(\hat{y}_1, \ldots, \hat{y}_M\right). \tag{20}$$

Here we use NeuroSED in equation 5 as the PRED function.

With multi-head pruning our model is able to explore multiple ways of pruning $G_t$ and thus has chances to capture good ones. At the same time, we only compute multiple prunings from different projections of the same set of node representations, so we only slightly increase the number of parameters.

### 4.3 Optimization

Prune4SED is summarized in Algorithm 1. First, query-aware representation learning encodes target graph $G_t$ (Line 1-5). Then multi-head pruning is used to prune $G_t$ from multiple views to predict SED (Line 6-9). Each head contains a sequence of operations, including $\boldsymbol{\alpha}$ computation, hard/soft pruning, and SED prediction. Finally, predictions from multi-head pruning are combined as the final prediction (Line 10).

For model training, model predictions are compared against SED values computed by a MIP-F2 solver (Lerouge et al., 2017). Given a pair of graphs $(G_t, G_q)$, the model makes a prediction $\hat{y}$. For the same pair the solver returns a lower bound $y_L$ and an upper bound $y_U$ of the true solution. In most cases $y_L = y_U$, which means that the solver find the true solution. For the same pair. To handle the general case, we penalize

the prediction with a squared error when it is outside of $[y_L, y_U]$. We minimize the following training loss computed from a dataset $\mathcal{D}$ to train the model.

$$\min \quad \mathcal{L} = \sum_{(G_q, G_t) \in \mathcal{D}} ([y_L - \hat{y}]_+)^2 + ([\hat{y} - y_U]_+)^2 \tag{21}$$

Here $[\cdot]_+$ truncate negative values to zero. The loss is equivalent to squared loss when $y_L = y_U$.

Next we analyze time complexity of Prune4SED. We first consider graph sizes. The attention layer in the QAL block needs to consider every node in $G_t$ against every node in $G_q$, so its running time is $O(|V_t| \cdot |V_q|)$. GAT layers runs in linear time about $O(|E_t| + |E_q|)$. Processing node vectors need time $O(|V_t| + |V_q|)$, so the overall running time is $O(|E_t| + |E_q| + |V_t| \cdot |V_q|)$.

The running time is in linear time with network hyperparameters, including the width of the network, the number of layers, the number of attention heads, or the number of pruning heads.

## 5 Experiments

In this section, we aim to validate the effectiveness of Prune4SED through experiments and also examine the model to understand how it works. We have the following sub-aims: 1) benchmarking the performance of Prune4SED against existing SED solvers/predicting models; 2) understanding the usefulness of our model designs through ablation studies; 3) examining pruned graphs to check whether neural graph pruning aligns with SED calculation; and 4) applying the model to a real-world applications.

**Experiment settings.** By default, we use $L = 5$ stages for Prune4SED. In hard pruning, we take top $k$ ($k = 5$) important nodes and their $h$ ($h = L - 1 = 4$) hop neighbors. The SED predictor contains an 8-layer GIN with 64 hidden units at every layer. We use $M = 5$ heads to produce the final prediction. More details about model hyperparameters and platforms are given in B.2.

**Datasets.** We use seven datasets (AIDS, CiteSeer, Cora_ML, Amazon, DBLP, PubMed, and Protein) to evaluate our model for SED approximation. B.1 provides more descriptions about these datasets.

We extract query-target pairs from each dataset to train, validate, and test models. For datasets with a single network (CiteSeer, Cora_ML, Amazon, DBLP, and PubMed), the target graph $G_t$ is a ego-net (up to 5-hop) of a random node sampled from large network. For datasets with multiple graphs (AIDS and Protein), the target graph is a graph in the dataset. Except the AIDS datset, the query graph is a subgraph from a random target graph (sampled from all target graphs). The subgraph is randomly sampled by starting from the center node of the target $G_t$, progressively selecting unseen neighbors with a probability 0.5 of the current nodes, and stopping at a depth up to 5. Query graphs of the AIDS dataset are known functional groups from Ranu & Singh (2009). To compute the SED between $G_q$ and $G_t$, we use a mixed-

Table 1: Dataset Statistics.

| **Dataset** | $|V_q|$ | $|V_t|$ | $|\Sigma|$ |
|---|---|---|---|
| AIDS | 9 | 14 | 38 |
| CiteSeer | 12 | 73 | 6 |
| Cora_ML | 11 | 98 | 7 |
| Amazon | 12 | 43 | 1 |
| DBLP | 14 | 240 | 8 |
| PubMed | 12 | 60 | 3 |
| Protein | 9 | 38 | 3 |

integer-programming solver, MIP-F2 (Lerouge et al., 2017). The solver runs on a 64 core machine with maximum of 60 seconds to compute lower and upper bounds the true SED. From each dataset, we randomly pair target and query graphs to get 100K pairs for training, 10K for validation, and another 10K for testing. Table 1 shows average sizes (rounded to 1) of target and query graphs and the number of node labels in the data extracted from each dataset.

**Baselines.** We compare Prune4SED with five other approaches. The first three are neural models: $H^2MN$ (Zhang et al., 2021), SIMGNN (Bai et al., 2019), and NeuroSED (Ranjan et al., 2021). For $H^2MN$ and SIMGNN, we use their modified versions from Ranjan et al. (2021) that better suit SED prediction. For $H^2MN$, we include its two versions of random walks ($H^2MN$-RW) and the k-hop version ($H^2MN$-NE). The second comparison contains two non-neural methods MIP-F2 (Lee et al., 2010a) and BRANCH (Blumenthal, 2019), both of which are implemented by an efficient C++ library GEDLIB (Blumenthal et al., 2019; 2020). As competing methods, both are given 0.1 second to compute the upper bound $y_U$ and lower bounds $y_L$ of

Table 2: RMSE on seven datasets.

|  | AIDS | CiteSeer | Cora_ML | Amazon | Dblp | PubMed | Protein |
|---|---|---|---|---|---|---|---|
| Branch | 1.379 | 3.161 | 3.102 | 4.513 | 2.917 | 2.613 | 2.391 |
| MIP-F2 | 1.537 | 4.474 | 3.871 | 5.595 | 3.427 | 3.399 | 2.249 |
| H$^2$MN-RW | 0.749 | 1.502 | 1.446 | 1.294 | 1.47 | 1.213 | 0.941 |
| H$^2$MN-NE | 0.657 | 1.827 | 1.229 | 0.971 | 1.552 | 1.326 | 0.755 |
| SIMGNN | 0.696 | 1.781 | 1.289 | 2.81 | 1.482 | 1.322 | 1.223 |
| NeuroSED | 0.512 | 0.519 | 0.635 | 0.495 | 0.964 | 0.728 | 0.524 |
| Prune4SED | **0.480** | **0.365** | **0.437** | **0.322** | **0.859** | **0.447** | **0.485** |
| Improv. over best baseline | 6.3% | 29.7% | 31.2% | 34.9% | 10.9% | 38.6% | 7.4% |
| Prune4SED *w/o.* QAL | 0.496 | 0.439 | 0.575 | 0.359 | 0.953 | 0.585 | 0.505 |
| Prune4SED *w.* 1 head | 0.497 | 0.383 | 0.458 | 0.324 | 0.915 | 0.469 | 0.495 |
| Prune4SED *w.* 10 heads | 0.482 | 0.373 | 0.449 | 0.321 | 0.817 | 0.488 | 0.488 |

the true SED, then $\hat{y} = (y_U + y_L)/2$ is used as the prediction. Note that 0.1 second is already much longer than our model's predicting time.

The predictive performance of these models is evaluated by root-mean-square error (RMSE), which is the square root of the mean squared error defined in equation 21.

## 5.1 Benchmark Predictive Performances

Table 2 summarizes performances of different models. The results of baselines are taken from Ranjan et al. (2021). The results show that neural methods outperform non-neural methods (Branch and MIP-F2). This is consistent with findings in previous work. H2MN and SIMGNN are designed for Graph Edit Distance. They compare the entire target graph to the query to compute the target value. This calculation is not appropriate for SED because many nodes in the target graph are irrelevant to the true SED.

Prune4SED outperforms all baselines and establishes new state-of-the-art performances across all seven datasets in terms of RMSE, as shown in Table 2. Specifically, Prune4SED substantially improves the best neural solver, NeuroSED, with an average of 23% improvement. Our analysis later shows that our model gains an advantage by removing some irrelevant nodes from the target graph before predicting SED values.

We further study the performance of our model with graph sizes. By controlling the random sampling procedure, we get graph pairs with different sizes and form three sets (small, medium, and large) of graph pairs. Each set has 2000 query-target pairs. We separately extract such sets from Cora_ML and Citeseer datasets. In the three sets (small/medium/large), the target graphs extracted from CiteSeer have average sizes 16/62/122, and those extracted from Cora have average sizes 17/64/118. The average sizes of queries are about 12 across all sets extracted from the two datasets. Each set are split into training (80%), validation (10%), and testing (10%). We run our model on each set to get an RMSE, then we can plot RMSEs from three sets against their sizes. We also run NeuroSED as a baseline here for a comparison.

Figure 4 shows the results. We see that both models have worse performances on larger graphs, which indicates that SED problems on larger graphs are more challenging for neural models. Nevertheless, the RMSE of our model is less than 1, which is still fairly accurate, considering that query graphs often have dozens of edges. Our model is still significantly better than NeuroSED.

We also test the predicting time of all models and report the comparison in Table 3. The optimization-based approach for

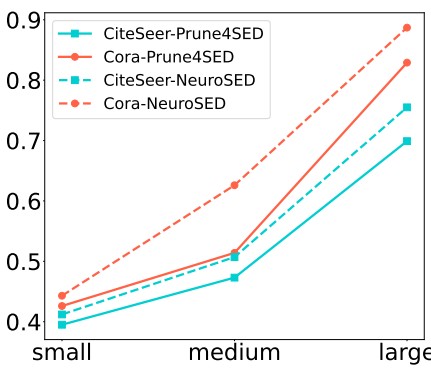

Figure 4: RMSE versus node number

Table 3: Inference time (seconds).

| **Dataset** | Cora_ML | CiteSeer |
|---|---|---|
| MIP-F2 | $1.0 \times 10^{-1}$ | $1.0 \times 10^{-1}$ |
| NeuroSED | $1.5 \times 10^{-4}$ | $1.5 \times 10^{-4}$ |
| Prune4SED | $1.9 \times 10^{-3}$ | $1.7 \times 10^{-3}$ |

SED calculation can take very long time to get the true answer. Here we give a time limit of 0.1 second to MIP-F2, and the time is already much longer than the prediction time of neural methods, but its performance is worse than neural methods as we have shown above. Prune4SED is slower than NeuroSED because it uses more neural layers. But considering the performance improvement, our model still has advantages in many applications. Furthermore, the computation time is not relevant to graph sizes and thus can be sped up by improving network architectures. We leave such work to the future.

## 5.2 Examine pruned graphs

To understand the good performance of the proposed model, we examine how the model prunes target graphs. We visualize a few examples of hard and soft pruning over the graph in Figure 5. Here node colors represent node labels. Nodes removed by hard pruning are not shown in the figure, and node sizes are proportional to keep probabilities.

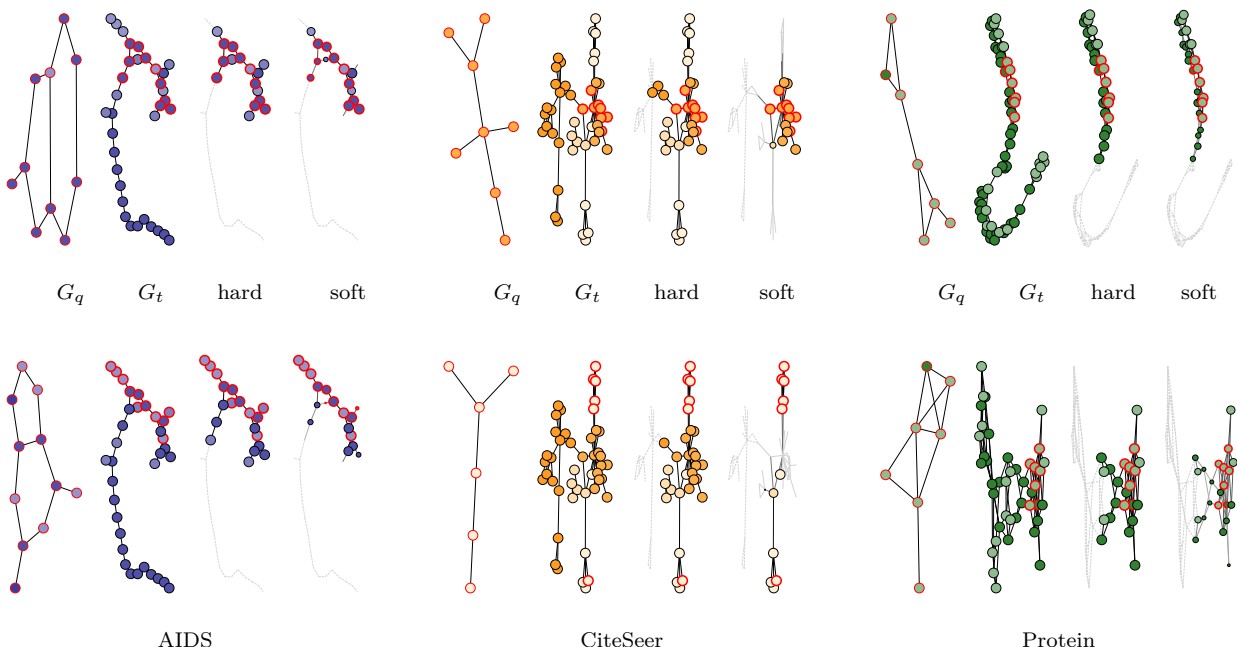

Figure 5: Visualization of pruned graphs on AIDS, CiteSeer, and Proteins datasets. Every four graphs is a group, representing query graph $G_q$, target graph $G_t$, target graph after hard pruning and after soft pruning. Node colors represent node labels. In soft pruning, nodes are resized according to their keep probabilities. Nodes with red circles mark the optimal subgraph.

These examples show that the pruning component removes a significant amount of unrelated nodes in the target graph. Hard and soft prunings obtained from $\boldsymbol{\alpha}$ are reasonable for matching the query. In the first example, it removes the long-chain, which obviously cannot match the query graph. We also observe that the pruning of the same target graph is different for different query graphs. In the AIDS example, nodes at the top part of the target graph are pruned differently; In the CiteSeer example, the nodes on the right side are pruned differently. It is a strong indication that the model does consider the query graph when it prunes the target graph.

We further check the effect of graph pruning over SED calculations. Ideally graph pruning should not affect SED calculation, that is, SEDs computed from pruned graphs should still be the same as SEDs computed from original target graphs. We truncate keep probabilities with a threshold and then prune all nodes with probabilities below the threshold, then we run an exact SED solver to check how much this hard way of

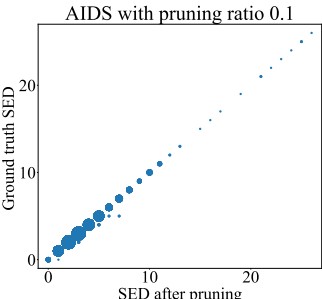 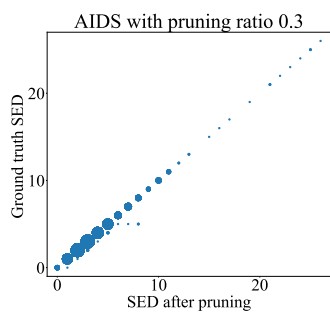 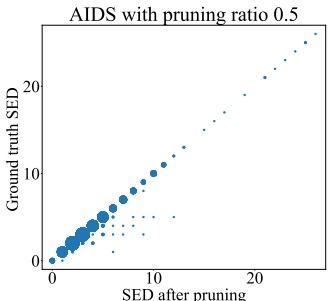

Figure 6: Correlation between SEDs computed from pruned graphs and SEDs computed from the original graphs. The size of dots indicates the number of SED pairs occurring at that location.

Table 4: Pearson correlation coefficient $\rho$ and mean absolute error (MAE) computed between ground truth SED and SED computed from pruned graph using an exact solver. The pruning ratio $r$ indicates the average node ratio impacted by hard pruning and soft pruning per graph. Larger results are better for $\rho$ and smaller results are better for MAE.

| **Dataset** | $r = 0.1$ | | $r = 0.3$ | | $r = 0.5$ | |
|---|---|---|---|---|---|---|
| | $\rho \uparrow$ | MAE $\downarrow$ | $\rho \uparrow$ | MAE $\downarrow$ | $\rho \uparrow$ | MAE $\downarrow$ |
| AIDS | 0.999 | 0.026 | 0.998 | 0.036 | 0.983 | 0.144 |
| CiteSeer | 1.000 | 0.022 | 1.000 | 0.026 | 1.000 | 0.032 |
| Protein | 0.996 | 0.092 | 0.995 | 0.110 | 0.980 | 0.237 |

pruning affects the SED. If the SED does not change much, then it means that node probabilities do indicate a good pruning of the target graph.

We conduct this experiment with 500 randomly selected graphs from the AIDS, CiteSeer, and Protein datasets. We adjust the threshold to prune $r = 0.1$, 0.3, and 0.5 of all graph nodes. Then we compare the calculated SEDs against SEDs computed from the original graphs (ground truth). The solver cannot get the true SED for a small number of cases, then we use $(y_U + y_L)/2$ as the true SED. We use MAE to measure the difference and also compute the correlation coefficient between the two groups of SEDs.

Table 4 summarizes the results of MAE and correlation coefficient. We see that the difference is fairly small when the ratio is small. When the pruning ratio is low (0.1 and 0.3), SEDs computed from pruned graphs only have slight differences with SEDs computed from the original graph. The average difference is less than 0.04 on the first two datasets and only 0.11 on the third dataset; the Pearson correlation coefficient is above 0.99. When the pruning ratio is as aggressive as 0.5, the difference is still at an acceptable level. Figure 6 further visualizes the correlation between SEDs computed from original graphs and pruned graphs respectively. The size of dots in the figure indicate the number of graph pairs occurring at that location in the plot. We again see that pruning does not affect SED computation on most graphs. These results confirm that our pruning weights correctly recognize and retain important nodes in $G_t$.

## 5.3 Ablation Studies

We conduct two ablation studies to investigate the effectiveness of the QAL block and multi-head pruning, and we report results in the bottom three rows 2.

**Query-aware representation learning.** Recall that query-aware learning aims to leverage query graph to learn representations for the target graph so that neural graph pruning is adaptive to the query graph.

To this end, we compare Prune4SED with and without query-aware learning. In the setup of Prune4SED without query-aware learning, we remove all QAL blocks in representation learning (i.e., $\mathbf{H}_t^l = \hat{\mathbf{H}}_t^l$ in equation 12 and equation 13 ). Therefore, representation learning solely depends on the target graph.

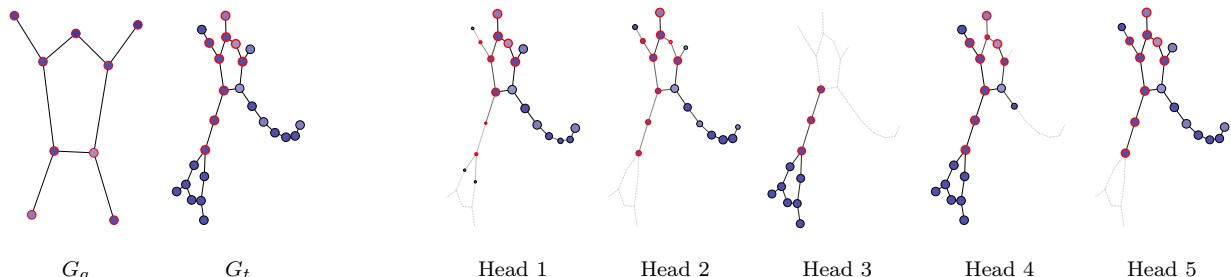

Figure 7: Multiple heads have learned different pruning strategies. By average, the final prediction obtained from multiple heads is close to ground truth (4.1 v.s. 4).

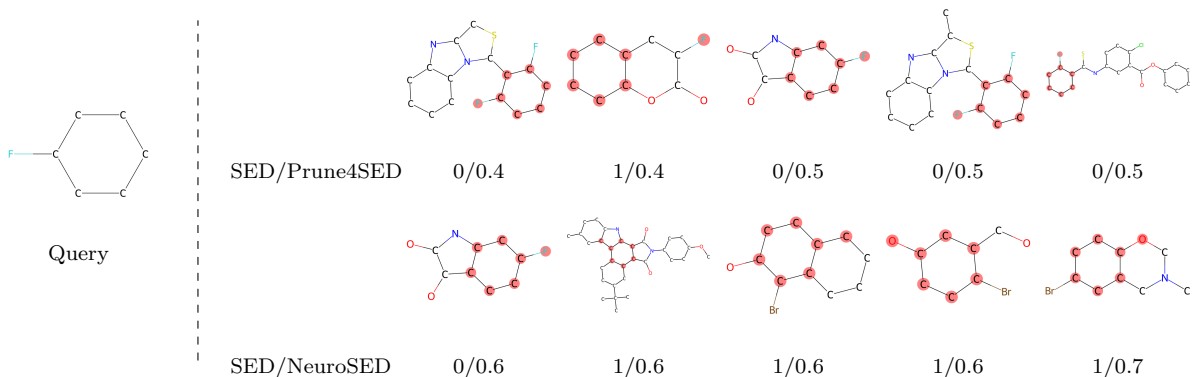

Figure 8: Retrieving chemical compounds given a query graph, which is a functional group. The retrieved compounds are ranked according to their predicted SED. Two values under each compound are ground truth SED and predicted SED.

We see that removing QAL blocks consistently increases RMSE values across seven datasets. The results verify the effectiveness of query-aware learning in terms of extracting information from query graphs.

**Multi-head pruning.** We use $M = 1$, $M = 5$ (default), and $M = 10$ heads to Prune4SED and compare their performances. Prune4SED with multiple heads ($M = 5$ and $M = 10$) clearly improves the performance compared with Prune4SED with a single predictive head, confirming the contribution of multiple heads. The results also suggest that $M = 5$ is a reasonable choice in practice.

Figure 7 further show an example where five predictive heads prune the target $G_t$ in different ways. Multiple predictive heads allow the model to discover more structures that are similar to the query graph and smooth out some predictive errors. The final prediction (averaged over 5 heads) is close to the ground truth (ground truth SED is 4, and the predicted SED is 4.1).

## 5.4 Application: Molecular Fragment Containment Search

Molecular fragment containment searching is a fundamental problem in drug discovery (Ranu et al., 2011; Ranjan et al., 2021). The problem can be formulated as a standard problem of subgraph similarity search: given a query graph representing some functional group, the task is to retrieve from a database chemical compounds that contain the functional group.

We apply Prune4SED to solve this task. We evaluate on AIDS dataset, where target graphs are antivirus screen chemical compounds collected from NCI [2] and query graphs are known functional groups. The dataset does not include Hydrogen atoms.

Figure 8 visualizes one set of retrieval results. The five molecules with the smallest predicted SEDs are shown in the Figure. Four out of these five molecules actually contain the function group. Prune4SED has better retrieval results than NeuroSED. More examples can be found in Figure 12 of B.4.

We also observe that Prune4SED can find molecules that contain the query but is much larger than the query. We hypothesize that our model is able to prune unnecessary nodes and identify the subgraph corresponding to the query, while NeuroSED either tries to find target graphs that are similar to the query graph or bet its luck on very large molecules. It is another evidence that pruning is necessary for SED calculation.

## 6 Conclusion

In this work, we study the problem of using a learning model to predict the SED between a target graph and a query graph. We present Prune4SED, an end-to-end model to address the problem. Notably, Prune4SED learns to prune the target graph to reduce interference from irrelevant nodes. It converts graph pruning to a node relabeling problem and enables pruning in a differentiable neural model. Our new model Prune4SED combines two novel techniques, query-aware representation learning and multi-head pruning to improve the model's ability of learning good prunings. Extensive experiments validate the superiority of Prune4SED.

### Acknowledgments

We thank all reviewers and editors for their insightful comments. Linfeng Liu and Li-Ping Liu were supported by NSF 1908617. Xu Han was supported by Tufts RA support. Part of the computation resource is provided by Amazon AWS.

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
