# OpenReview forum: "Towards Accurate Subgraph Similarity Computation via Neural Graph Pruning"
_TMLR — Accepted by TMLR_

### Review · Reviewer_LCjD · 2022-07-12

**Summary Of Contributions:**

This paper investigates the problem of subgraph similarity computation via neural graph pruning. Subgraph similarity computation is a hot topic in the field of graph analysis, and has many applications in real-world scenarios. To address this problem, the authors start from converting graph pruning to a problem of node relabeling, then relax it to a differentiable problem. Therefore, the authors then propose a neural network for SED based on this point. Experiments on several benchmarks demonstrate the effectiveness of this proposed model.

**Broader Impact Concerns:**

None.

**Requested Changes:**

Please see the weaknesses in Strengths And Weaknesses.

**Strengths And Weaknesses:**


Strengths:

1. The paper is well-written and easy to follow.
2. Starting from a traditional problem, namely a non-differentiable labeling problem, and relaxing to a differentiable neural network, I think is a good direction for problem analysis.



Weaknesses:

1. The related studies are not quite sufficient. Some related studies are not cited and well-discussed, such as the subgraph isomorphism counting approaches [1,2].

2. The proposed neural network model is straightforward and simple.

3. It seems that the employed datasets are not large, and only a few query and target graphs are utilized, which may give rise to unconvincing results. Besides, the division of datasets is not clear.

4. The analysis for the experiments is not sufficient. Only the observations are given, but the insights that why the model can outperform the baselines (or other observations) are not well-illustrated. It is the same case for the other experimental results, such as the ablation study.


[1] neural subgraph isomorphism counting. KDD 2020
[2] Can graph neural networks count substructures? NeurIPS 2020

---

> ### Author Response · Authors · 2022-07-17
> **Thank you for your feedback!**
>
> We sincerely appreciate your constructive feedback. We'd like to address your concerns as follows.
>
> **Q1**: Some related studies are not cited and well-discussed, such as the subgraph isomorphism counting approaches.
>
> *Response:* Thank you for pointing out these references. We will put them in the next version and discuss the relation as follows. Both [1] and [2] study the subgraph isomorphism counting problem, which requires the model to match a graph pattern to possibly multiple subgraphs in a target graph. Subgraph similarity is a relevant but somewhat different problem: it needs to find the best (not necessarily exact) match between the query graph and a subgraph of the target.
>
> **Q2**: The proposed neural network model is straightforward and simple.
>
> *Response:* We would like to point out that we did try various architectures and eventually found the architecture in the submission works the best, so our model design does carry our thoughts. We'd like to highlight three points here: (1) the Query-Aware Learning (QAL) block extracts information about the query graph to guide the pruning of the target graph; (2) the hard pruning can quickly reduce the size of the target graph while the soft pruning keeps differentiability; and (3) the multi-head prediction strategy can explore multiple ways of pruning and thus increase the chance of capturing good ones.
>
> Considering its premium performance in real tasks, even if the architecture is easy to grasp, we don't feel that is a weakness.
>
> We hope we have correctly understood the question. Please clarify if our response does not address your concern.
>
>
> **Q3**: Datasets are not large, and only a few query and target graphs are utilized. The division of datasets is not clear.
>
> *Response:* Thanks for the comment. In Section 5 (below Table 1), we have a description of the data split. From each dataset, we use 100,000 random query-target pairs for training, 10,000 random pairs for validation, and another 10,000 random pairs for testing. The datasets should be large enough to evaluate the performance of the comparison methods. We will put it into a separate paragraph to make it more visible.
>
> **Q4**: The analysis for the experiments is not sufficient. Only the observations are given, but the insights that why the model can outperform the baselines (or other observations) are not well-illustrated. It is the same case for the other experimental results, such as the ablation study.
>
> *Response:*  Thanks for the feedback. The results in Table 2 show that neural methods outperform non-neural methods (Branch and MIP-F2). This is consistent with findings in previous work. H2MN and SIMGNN are designed for Graph Edit Distance. They compare the entire target graph to the query to compute the target value. This calculation is not appropriate for SED because many nodes in the target graph are irrelevant to the true SED.
>
> Our main competitor is NeuroSED. The advantage of our model is that the pruning component can remove some irrelevant nodes from the target graph before the final calculation of SED. Examples in Figure 4 illustrate this point through several examples. Figure 7 shows that our pruning does not hurt the ground-truth calculation of SED. Our ablation study shows the usefulness of incorporating the query graph when learning the representation of the target graph.
>
> We hope our response above clarifies your concern. Please do let us know if there are still questions.

---

> > ### Author Response · Authors · 2022-07-22
> > **One extra note**
> >
> > On question 4: In the next version, we will add more analysis of experiment results. The discussion will adapt our answer to your Q4 and also provide a more detailed explanation of why the model outperforms baselines.
> >
> > Thanks again for the feedback!

---

### Review · Reviewer_qdtj · 2022-07-22

**Summary Of Contributions:**

The main contribution of the paper is a model that learns to extract (from a target graph) a subgraph that is similar to a query graph. The considered problem generalizes the subgraph isomorphism problem and is thus hard. The proposed (heuristic) approach uses a neural network to prune nodes that are potentially not of interest, and then another component of the model predicts the subgraph edit distance. The proposed model is evaluated on several benchmark datasets where it outperforms the baselines.

**Broader Impact Concerns:**

A Broader Impact Statement  is not required.

**Requested Changes:**

- The authors should discuss in more details why hard pruning is necessary and how exactly training is performed (since hard pruning is not differentiable in my understanding). Furthermore, in case hard pruning is performed, what does soft pruning further offer? I would expect the model to perform either hard or soft pruning, but not both. The authors should provide clear explanations on that.

- Since the neighborhood aggregation approach (GATv2) is one of the main components of Prune4SED, and Prune4SED's performance depends a lot on it, I would like to see some more experiments where some more expressive neighborhood aggregation approach is employed.

- The problem becomes more challenging when the size (number of nodes) of both the target graph and the query graph increases. The average size of the graphs in the different datasets is not very large in most cases. I suggest the authors also experiment with some larger graphs (both query and target) such that it becomes clear what is the impact of graph size on the model's performance.

- The authors should also report the running time of the model (both training and inference time) since this is the main advantage of the proposed model compared to existing solvers. I suggest they also compare the running time of the proposed Prune4SED model against those of the baselines.

- Algorithm 1 has some issues. For instance, in line 7, it is not clear from the previous lines what does exactly $\mathbf{H}_t$ represent. In addition, I think that in line 11, $y$ should be replaced with $\hat{y}$.

- There are some typos that the authors should fix. For example:\
p.3: "PRELIMINARIES" -> "Preliminaries"\
p.7: "each of which optimize" -> "each of which optimizes"\
p.7: "Then they get" -> "Then we get"\
p.7: "in equation equation" -> "in equation"\
p.8: "datset, The" -> "dataset, the"

**Strengths And Weaknesses:**

Strengths
--
- Overall, in my view, the paper is interesting. The problem of discovering one or more subgraphs that are as similar as possible to a query graph is essential for many applications and this paper is one of the first steps towards this direction (in the field of graph representation learning).

- The proposed model seems to be novel. To be precise, the different components of the model are not entirely novel since they have been introduced/employed in previous works, however, in my opinion, combining those components to deal with the subgraph search task is not so trivial.

- The proposed model establishes a new state-of-the-art in the subgraph search task. On some datasets, Prune4SED provides significant improvements, while the experiments reported in Table 3 indicate that the pruning component indeed removes from the target graph mainly nodes that are not of any interest.

Weaknesses
--
- The main contribution of the paper (and its main selling point) is that it provides a differentiable pruning component. However, the authors apply hard pruning to remove most of the nodes (they keep $k=5$ nodes and their neighborhoods). First of all, it is not clear to me how the proposed model is end-to-end trainable since the hard pruning is not differentiable in my understanding.

- The authors also fail to give proper explanations on why both hard and soft pruning are applied and what is the exact purpose of soft pruning.

- To update the representations of the nodes of both target and query graphs, the authors employ the GATv2 model. GAT is less expressive that 1-WL in terms of distinguishing non-isomorphic graphs (I guess the same is true for GATv2). This is a major limitation of the proposed model since it will fail to retrieve subgraphs of good quality in case of symmetries. I know that in most real-word graphs, this is not the case, however, it would strengthen the approach if more expressive node representations were learned.

---

> ### Author Response · Authors · 2022-07-29
> **Thank you for your feedback!**
>
> Thank you for the insightful feedback. We would like to answer your questions and clarify a few points below.
>
> **Q1**: It is not clear to me how the proposed model is end-to-end trainable...  why both hard and soft pruning are applied and what is the exact purpose of soft pruning.
>
> *Response*:  Let's first clarify how the model works. The model runs both hard pruning and soft pruning *at the same time*: hard pruning removes nodes with small probabilities (eq 16&17), and soft pruning multiplies probabilities to remaining nodes (eq 18).  We directly train the *entire* model by minimizing the loss. This structure is similar to Graph U-Net [8].
>
> Soft pruning is a differentiable operation, with which the model learns to use zero probabilities to indicate unmatching nodes in the target graph. Then hard pruning can remove nodes with small keep probabilities because these nodes have little hope to match nodes in the query graph.
>
> We agree that hard pruning is not differentiable and will make this clear in the next version. We use the term "end-to-end" to highlight that the model can directly predict SED without any intervention during intermediate steps. It is consistent with the  "end-to-end deep learning" in the Coursera course [6] taught by Andrew Ng. Is it acceptable to call our network an "end-to-end learning model" but with a "non-differentiable" operation?
>
> We will add discussion to make this point clearer.
>
> **Q2**: GAT is less expressive that 1-WL in terms of distinguishing non-isomorphic graphs
>
> *Response*: It is a great point to strengthen our approach with more expressive GNNs. We considered one approach of adding a random feature to each node [2] to improve the expressiveness. We tried this method but did not see much accuracy gain. Specifically, with GNNs and GNNs using random features, the RMSE results are (0.480 vs 0.480) on AIDS, (0.365 vs 0.368) on CiteSeer, and (0.485 vs 0.486) on Protein. Maybe the symmetry does not cause much trouble in our case. We will add a discussion in our new version.
>
> **Q3**: The problem becomes more challenging when the size (number of nodes) of both the target graph and the query graph increases...
>
> *Response*: Thanks for the comments. Computing (approximate) SED on large target/query graphs is still challenging for both non-neural and neural methods. When a target graph is large, the hard pruning operation can remove a large fraction of irrelevant nodes. However, it is good to know where the limit is. When the query graph is large, the problem is hard by itself.  However, our model still directly benefits quite a few real-world applications (e.g., in drug discovery [3] and WikiData graph [4])  with small query graphs. We will add these discussions to the submission in the next version.
>
> Currently, we are running new experiments on larger graphs. We will report our results shortly.
>
>
> **Q4**: The authors should also report the running time of the model...
>
> Response: Thanks for the feedback. The results in Section 5.1 shows the fast inference time of our model. To compute SED for a pair of target and query, we give 0.1 seconds for a non-neural method MIP-F2 [7]. Prune4SED runs 100x faster than MIP-F2: on average it takes only 0.001 seconds. Our model has similar training and inference time compared to the state-of-the-art neural method [5].
>
> **Q5**: Algorithm 1 has some issues...
>
> *Response*:  $H^t$ in line 7 is the concatenation of $H_t^l$ generated from line 4; line 7 corresponds to equation (14). For line 11, we agree that $y$ should be replaced with $\hat{y}$. We will add the calculation of $H^t$ to Algorithm 1 and fix the typo.
>
> **Q6**: Fix typos.
>
> *Response*: We thank the reviewer for pointing out these typos. We will address these typos in our new version.
>
>
> **References**:
>
> [1] Glasmachers, Tobias. "Limits of end-to-end learning." Asian conference on machine learning. PMLR, 2017.
>
> [2] Sato, Ryoma, Makoto Yamada, and Hisashi Kashima. "Random features strengthen graph neural networks." SIAM 2021.
>
> [3] Ranu, Sayan, and Ambuj K. Singh. "Mining statistically significant molecular substructures for efficient molecular classification." *Journal of chemical information and modeling* 2009.
>
> [4] Bonifati, Angela, Wim Martens, and Thomas Timm. "An analytical study of large SPARQL query logs." *The VLDB Journal* 2020).
>
> [5] Ranjan, Rishabh, et al. "A Neural Framework for Learning Subgraph and Graph Similarity Measures." arXiv 2021.
>
> [6] Andrew Ng. What is End-to-end Deep Learning? Coursera. Link: https://www.coursera.org/lecture/machine-learning-projects/what-is-end-to-end-deep-learning-k0Klk
>
> [7] Lee, Jungmin, Minsu Cho, and Kyoung Mu Lee. "A graph matching algorithm using data-driven markov chain monte carlo sampling." *2010 20th International Conference on Pattern Recognition*. IEEE, 2010.
>
> [8] Gao, Hongyang, and Shuiwang Ji. "Graph u-nets." international conference on machine learning. PMLR, 2019.

---

### Review · Reviewer_aL42 · 2022-08-03

**Summary Of Contributions:**

This submission studies the important question of subgraph similarity matching. The submission proposes to convert the subgraph matching problem to a graph pruning problem, which is instantiated as a node labeling problem (labeling nodes as keep or pruned). Such problem formulation enables end-to-end differentiable learning, which motivates the usage of neural methods. The submission also designs special architecture for this application. Namely, the authors propose a query-aware graph encoding network and proposes to use multi-head prediction for the node relabeling problem.

This work evaluates on seven datasets to study the problem of subgraph similarity search. The datasets are created by automatically converting existing graph benchmarks. The proposed method is compared to seven prior baselines and demonstrated superior performance on all seven benchmarks.

Overall, I think this submission studies an interesting problem in practice, proposes a novel perspective that converts the similarity search into a pruning problem, and proposes an empirically strong method to solve this. I enjoyed reading the submission and would recommend acceptance.

**Broader Impact Concerns:**

I do not have ethical concerns with the submission

**Requested Changes:**

Some comments on writing:
1. Page 16, under section A.2 equation number is missing

**Strengths And Weaknesses:**

Strengths:
1. the paper studies an interesting question with clear motivation
2. the proposed method is novel
3. the proposed method is empirically strong compared to prior baselines
4. the submission carries out detailed ablation studies and qualitative studies to understand the proposed method

Weakness:
1. Minor concern: all the benchmarks used to evaluate this work is created automatically. Does such automatic transformation reflects the problems we see in real-world applications? Can the authors comment on this?

---

### Author Response · Authors · 2022-08-17
**Just in case you have last-minute questions**

According to a message we have received from TMLR, today is the last day we can respond to your questions. Besides posting answers to your comments, we have also updated our submission. Please check our comments and updates, and let us know if you have any questions. Thank you again for your comments.

---

### Decision · Action_Editors · 2022-09-16

**Recommendation:** Accept with minor revision

**Comment:**

This paper proposes a novel and interesting approach to compute approximate Subgraph Edit Distance using Graph neural networks.  The simultaneous smooth and hard pruning is interesting and works well in practice despite the loss of differentiability. The reviewers found the paper interesting and all believe it will be helpful to the community. But they also had several concerns about the paper.

Most of those concerns have been addressed in the discussion with reviewers and in the current version.  For these reasons the paper should be accepted but there remain several small but important points that must be integrated to the paper so we will require minor revisions.

Below are the expected revisions:
+ The discussion with the reviewers about inference time is interesting and state that it is provided in section 5.1 which is false. The authors need to add a an inference time table that compare the methods from a computational time perspective. This is necessary to show the interest of the method independently of performance.
+ The question about the quality of the method as a function of the size of the graphs is important. the authors did a new experiments on a smaller number of larger graphs that suggests that the method still works better than the state of the art competitor which should be integrated in the paper with a discussion. Also it might be interesting to see the effect of the size of the graphs (both query and target) on the performance for instance by plotting on a dataset a scatterplot of the performance VS size to see if there is indeed a loss in performance. In any case all this should be integrated in the paper instead of just in the comments.
+ The authors add the relative improvement of the method in parenthesis after their performance. This is a non standard way to show this and might be confused with variances that are traditionally given similarly. The authors should have a separate line in the tables clearly named to report the improvement of the method.

---

> ### Author Response · Authors · 2022-09-20
> **Thank you so much for the recommendation and the feedback!**
>
> We will address these concerns in our camera-ready version.

---

> > ### Author Response · Authors · 2022-10-19
> > **Camera-ready version submitted**
> >
> > Thanks again for the recommendation and the feedback. We have made the following revisions according to the comment in the camera-ready version.
> >
> > 1. We have provided inference time in Table 3. The comparison shows the performance of MIP-F2 is much worse than the performances of neural methods even if it uses a much longer time than neural methods. Our method runs slower than NeuroSED because we have more neural layers for pruning. Considering the performance advantage, our method is still preferable in many applications.
> >
> > 2. We include the plot of performances vs graph sizes. Our method still has a performance advantage over NeuroSED, though both methods have worse performance when graph sizes are larger. (We created scatter plots of errors vs graph sizes from the SAME dataset and the SAME trained model, but we did not find a clear pattern there. Maybe when target graph sizes are similar (e.g. around 16), the correlation is not very obvious. We tend to omit this discussion from this version)
> >
> > 3. We have put relative improvements in a separate row in Table 2. We hope this is clearer.
> >
> > Besides the suggested revisions, we have also made the following revisions.
> >
> > 4. We have revised the section on related work to emphasize the relation between previous work and this work.
> >
> > 5. We have made the problem definition of SED calculation clearer
> >
> > 6. We revise the loss calculation to include rare cases when we only have lower and upper bounds of SED
> >
> > 7. We have added more experiment details.
> >
> > 8. We have improved the arrangement of figures and tables so they are closer to their references.
> >
> > Please check our revisions and let us know if there are more changes you want to see. Thank you!